# Data-Driven Control Techniques for Renewable Energy Conversion Systems: Wind Turbine and Hydroelectric Plants

**Silvio Simani** *[ID]**, Stefano Alvisi**[ID] **and Mauro Venturini**[ID]

Dipartimento di Ingegneria, Università degli Studi di Ferrara, Via Saragat 1E, 44122 Ferrara (FE), Italy; stefano.alvisi@unife.it (S.A.); mauro.venturini@unife.it (M.V.)

* Correspondence: silvio.simani@unife.it; Tel.: +39-0532-97-4844

**Abstract:** The interest in the use of renewable energy resources is increasing, especially towards wind and hydro powers, which should be efficiently converted into electric energy via suitable technology tools. To this end, data-driven control techniques represent viable strategies that can be employed for this purpose, due to the features of these nonlinear dynamic processes of working over a wide range of operating conditions, driven by stochastic inputs, excitations and disturbances. Therefore, the paper aims at providing some guidelines on the design and the application of different data-driven control strategies to a wind turbine benchmark and a hydroelectric simulator. They rely on self-tuning PID, fuzzy logic, adaptive and model predictive control methodologies. Some of the considered methods, such as fuzzy and adaptive controllers, were successfully verified on wind turbine systems, and similar advantages may thus derive from their appropriate implementation and application to hydroelectric plants. These issues represent the key features of the work, which provides some details of the implementation of the proposed control strategies to these energy conversion systems. The simulations will highlight that the fuzzy regulators are able to provide good tracking capabilities, which are outperformed by adaptive and model predictive control schemes. The working conditions of the considered processes will be also taken into account in order to highlight the reliability and robustness characteristics of the developed control strategies, especially interesting for remote and relatively inaccessible location of many plants.

**Keywords:** wind turbine system; hydroelectric plant simulator; model-based control; data-driven approach; self-tuning control; robustness and reliability

---

## 1. Introduction

The trend to reduce the use of fossil fuels, motivated by the need to meet greenhouse gas emission limits, has driven much interest in renewable energy resources, in order also to cover global energy requirements. Wind turbine systems, which now represent a mature technology, have had much more development with respect to other energy conversion systems, e.g., for biomass, solar, and hydropower [1]. In particular, hydroelectric plants present interesting energy conversion potentials, with commonalities and contrast with respect to wind turbine installations [2–4].

One common aspect regarding the design of the renewable energy conversion system concerns the conversion efficiency. However, as wind and hydraulic resources are free, the focus is on the minimisation of the cost per kWh, also considering the lifetime of the plant. Moreover, by taking into account that the cost of the control system technology (i.e., sensors, actuators, computer, software) is relatively lower than the one of the renewable energy converter, the control system should aim at increasing the energy conversion capacity of the given plant [5].



The paper focuses on the development and the comparison of different control techniques applied to a wind turbine system and a hydroelectric plant, by using a wind turbine benchmark and a hydroelectric simulator, respectively. The former process was proposed for the purpose of an international competition started in 2009 [6], whilst the latter system was developed by the authors but with different aims [7]. In fact, these simulators represent high-fidelity representations of realistic processes, developed for the validation of fault diagnosis and fault tolerant control techniques [7,8]. More general investigations of these plants and their components are addressed in [9] and [10], respectively, even if their structures were analysed for different purposes and applications.

With reference to wind turbine systems, their regulation can be realised via 'passive' control methods, such as the plants with fixed-pitch and stall control machines. These systems may not use any pitch control mechanisms or they rely on simple rotational speed control [6]. On the other hand, wind turbine rotors exploiting adjustable pitch systems are often exploited to overcome the limitations due to the simple blade stall, and to improve the converted power [11]. Large wind turbines can implement another control technique modifying the yaw angle, which is thus used to orient the rotor towards the wind direction [11].

On the other hand, regarding hydroelectric plants, it is worth noting that a limited number of works addressed the application of advanced control techniques [12]. In fact, a high-fidelity mathematical description of these processes can be difficult to be achieved in practice. Some contributions took into account the elastic water effects, even if the nonlinear dynamics are linearised around an operating condition. Other papers proposed different mathematical models with the related control strategies [13]. To this end, linear and nonlinear dynamic processes with different regulation strategies are also considered [14]. In particular, a fuzzy controller that needs for the proper design of the membership functions was addressed in [15]. On the other hand, an advanced control logic combining four control schemes that rely on adaptive, fuzzy and neural network regulators was investigated in [13]. Finally, regarding joint wind-hydro deployments, some more recent works analysed the problem of frequency control of isolated systems [16,17].

After these considerations, the main contribution of the paper aims at providing some guidelines on the design and the application of data-driven and self-tuning control strategies to a wind turbine benchmark and a hydroelectric plant simulator. Some of these techniques were already applied to wind turbine systems, and important advantages may thus derive from the appropriate implementation of the same control methods in hydroelectric plants. In fact, investigations considering the control problem of both wind turbine systems and hydroelectric plants present a limited number of common points, thus leading to little exchange of shared features. This consideration is particularly valid with reference to the well established wind turbine area when compared to hydroelectric systems. Moreover, the work analyses the application of the different control solutions to these energy conversion systems. In particular, the paper introduces some kind of common rules for tuning the different controllers, for both the wind turbine system and the hydroelectric plant. Therefore, the paper shows that the parameters of these controllers are obtained by exploiting the same tuning strategies. This represents an important characteristic of this study. The common parts and the working conditions of these energy conversion systems will be also taken into account in order to highlight the reliability and robustness characteristics of the developed control strategies.

Finally, the paper has the following structure. Section 2 recalls the simulation models used for describing the accurate behaviour of the plants. In particular, similar functional parts that characterise the processes under investigation are highlighted, as they lead to similar design rules. To this end, Section 3 summarises the design of the proposed control techniques, taking into account the available tools. Section 4 shows the implementation of these control strategies, which are compared to the achievable reliability and robustness features. Section 5 ends the paper summarising the main achievements of the paper, and drawing some concluding remarks.

## 2. Simulator Models and Reference Governors

This section recalls the basic structure and the common functional modules of the simulators used for describing the wind turbine and the hydroelectric processes considered in this paper.

First, this work proposes a horizontal-axis wind turbine device, as nowadays it represents the most common type of solution for large-scale deployments. Moreover, this three-bladed wind turbine follows the principle that the wind power activates its blades, thus producing the rotation of the low speed rotor shaft. This rotational speed required by the electric generator is increased via a gear-box with a drive-train [6]. The schematic diagram of this benchmark that helps to recall its main variables and function blocks developed in the Simulink environment is depicted in Figure 1.

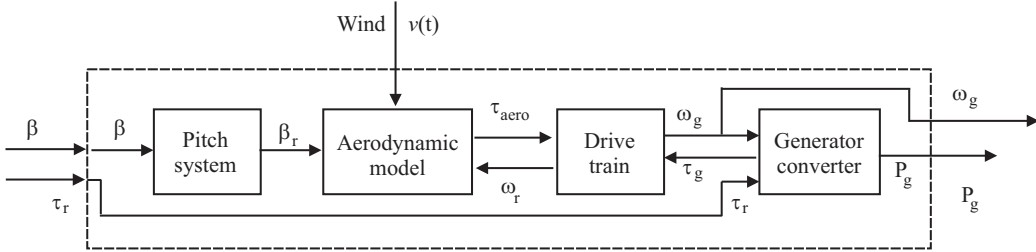

**Figure 1.** Block diagram of the wind turbine simulator.

The wind turbine simulator has two controlled outputs, i.e., the generator rotational speed $\omega_g(t)$ and its generated power $P_g(t)$. The wind turbine model is controlled by means of two actuated inputs, i.e., the generator torque $\tau_g(t)$ and the blade pitch angle $\beta(t)$. The latter signal controls the blade actuators, which are implemented by hydraulic circuits [6].

Several other measurements are acquired from the wind turbine benchmark. $\omega_r(t)$ represents the rotor speed and $\tau_r(t)$ is the reference torque. Moreover, the aerodynamic torque $\tau_{aero}(t)$ is computed from the wind speed $v(t)$, which is usually available with limited accuracy. Moreover, $\tau_{aero}(t)$ depends on the power coefficient $C_p$, as shown by the relation of Equation (1):

$$\tau_{aero}(t) = \frac{\rho \, A \, C_p \left( \beta(t), \, \lambda(t) \right) \, v^3(t)}{2 \, \omega_r(t)}, \tag{1}$$

$\rho$ being air density, $A$ the area swept by the turbine blades during their rotation, whilst $\lambda(t)$ is the tip-speed ratio. The nonlinear relations of Equation (1) is represented in Figure 2, which is also depicted for different values of $\beta$.

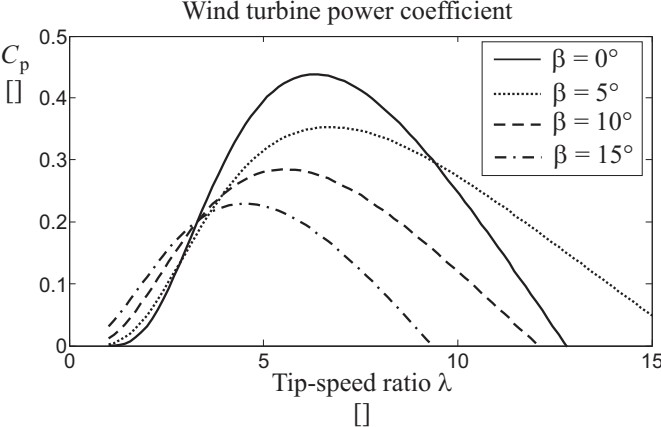

**Figure 2.** Example of power coefficient function $C_p(\beta, \lambda)$.

It is worth noting that the relation of Equation (1) representing the driving force of the wind turbine process has a similar formulation in hydroelectric plant model, as shown in the following.

The continuous-time model of the wind turbine benchmark can be described by the system of Equation (2):

$$
\begin{cases}
\dot{x}(t) & = & f_c\left(x(t), u(t)\right), \\
\\
y(t) & = & x(t),
\end{cases}
\tag{2}
$$

where $u(t) = [\tau_r(t)\,\beta(t)]^T$ and $y(t) = [\omega_g(t)\,P_g(t)]^T$ is the input vector. $f_c(\cdot)$ is described by means of a continuous-time nonlinear function representing the dynamic behaviour of the controlled process. Moreover, since this paper will analyse several data-driven control approaches, the system of Equation (2) will be used to acquire $N$ sampled data sequences $u(k)$ and $y(k)$, with $k = 1, 2, \ldots N$.

Finally, the wind turbine simulator includes a control scheme that maintains the generator speed $\omega_g(t)$ at its nominal value $\omega_{nom} = 1551.76$ rpm, and the generated power $P_g(t)$ close to the rated power $P_r = 4.8$ MW. This is achieved by properly actuating both $\beta$ and $\tau_g$, depending on the operating conditions, which move the wind turbine system from the partial load to the full load working regions (the operating regions 2 and 3, respectively) [6].

On the other hand, the hydroelectric plant considered in this work consists of a high water head and a long penstock, which also includes upstream and downstream surge tanks, with a Francis hydraulic turbine [18], as recalled in Figure 3. Therefore, the hydroelectric simulator consists of a reservoir with water level $H_R$, an upstream water tunnel with cross-section area $A_1$ and length $L_1$, an upstream surge tank with cross-section area $A_2$ and water level $H_2$ of appropriate dimensions. A downstream surge tank with cross-section area $A_4$ and water level $H_4$ follows, ending with a downstream tail water tunnel of cross-section area $A_5$ and length $L_5$. Moreover, between the Francis hydraulic turbine and the two surge tanks, there is a penstock with cross-section area $A_3$ and length $L_3$. Finally, Figure 3 highlights a tail water lake with level $H_T$. The levels $H_R$ and $H_T$ of the reservoir and the lake water, respectively, are assumed to be constants.

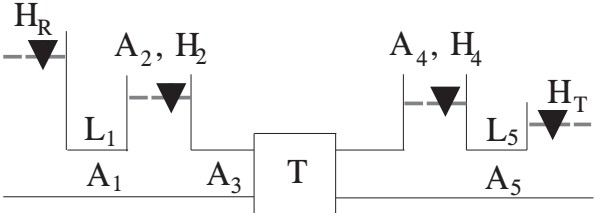

**Figure 3.** Scheme of the hydroelectric process.

The overall model of the hydroelectric simulator is described by the relations of Equation (3), which express the non-dimensional variables with respect to their relative deviations [7,19]:

$$
\begin{cases}
\frac{Q}{Q_r} & = & 1 + q_t, \\
\frac{H}{H_r} & = & 1 + h_t, \\
\frac{n}{n_r} & = & 1 + x, \\
G & = & 1 + y,
\end{cases}
\tag{3}
$$

where $q_t$ is the turbine flow rate relative deviation, $h_t$ the turbine water pressure relative deviation, $x$ the turbine speed relative deviation, and $y$ the wicket gate servomotor stroke relative deviation. In particular, in Equation (3), $H_r = 400$ m represents the reservoir water level, $Q_r = 36.13$ m$^3$/s is the water flow rate, and $n_r = 500$ rpm is the rated rotational speed. The hydraulic turbine power is $P_r = 127.6$ MW with rated efficiency $\eta_r = 0.90$.

In the following, the non-dimensional performance curves of the hydraulic turbine considered in this work are briefly summarised, as they represent an important nonlinear part of the hydroelectric plant. In particular, the non-dimensional water flow rate $Q/Q_r$ is expressed as a function of

the non-dimensional rotational speed $n/n_r$, and represented by the second order polynomial of Equation (4):

$$\frac{Q}{Q_r} = G \left[ a_1 \left( \frac{n}{n_r} \right)^2 + b_1 \left( \frac{n}{n_r} \right) + c_1 \right] = f_1\left(n, G\right). \tag{4}$$

Moreover, the relation of Equation (4) includes the wicked gate opening, described by the non-dimensional parameter $G$, varying from 0 to 100%. As an example, Figure 4 represents the function of Equation (4) for different values of the wicked gate opening $G$, which defines the operating conditions of the Francis hydraulic turbine.

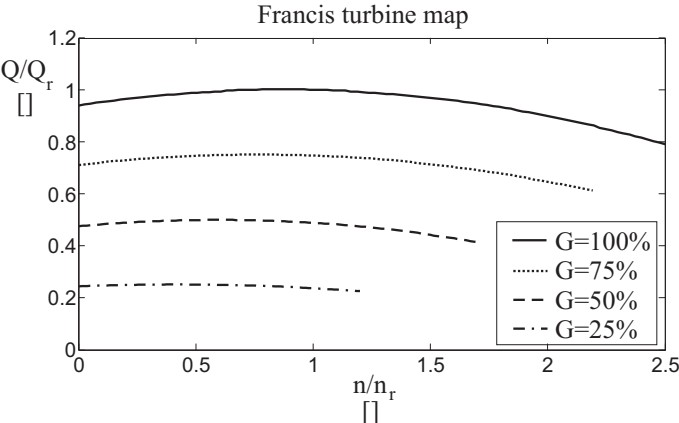

**Figure 4.** Francis turbine map for different values of $G$.

Note that the function of Equation (4) plays the same role of the curve represented by Equation (1).

The parameters of the hydroelectric model were selected in order to represent a realistic hydroelectric plant simulator [19]. Moreover, as for the wind turbine benchmark, the signals that can be acquired from the hydroelectric plant simulator are modelled as the sum of the actual variables and suitable stochastic processes [7]. For this benchmark, a standard PID regulator was proposed to compensate the hydraulic turbine speed [19]. Due to its nonlinear characteristics, this solution may lead to unsatisfactory responses, with high overshoot and long settling time, as highlighted in [19], since a gain scheduling of the PID parameters would have been required. Thus, advanced control strategies that were already proposed for the wind turbine benchmark and recalled in Section 3 will be briefly summarised and applied to the hydroelectric simulator, as shown in Section 4. Extended simulations, comparisons, and the sensitivity analysis of the proposed solutions represent one of the key points of this paper.

Finally, it is worth noting that some relations of the hydroelectric system have been linearised; see, e.g., the system of Equation (3). However, these linear approximations are performed so that the remaining nonlinear parts of the considered processes are closer, as highlighted by Equations (1) and (4).

## 3. Control Techniques for Energy Conversion Systems

This section describes briefly several control schemes consisting of self-tuning, data-driven, and Artificial Intelligence (AI) strategies, such as fuzzy logic and adaptive methods, as well as Model Predictive Control (MPC) approach. First, with reference to the process output, the desired transient or steady-state responses can be considered, as for the case of self-tuning PID regulators summarised in Section 3.1. On the other hand, if the frequency behaviour is taken into account, the desired closed-loop poles can be fixed as roots of the closed-loop transfer function. This represents the design approach used by the adaptive strategy considered in Section 3.3. Moreover, when robust performances are included, the minimisation of the sensitivity of the closed-loop system with respect to the model-reality mismatch or external disturbances can be considered. This approach is related to

the fuzzy logic methodology reported in Section 3.2. Some other strategies provide solutions to this optimisation problem when it is defined at each time step, as for the Model Predictive Control (MPC) with disturbance decoupling considered in Section 3.4. This strategy integrates the advantages of the MPC solution with the disturbance compensation feature.

### 3.1. Self-Tuning PID Control

Industrial processes commonly exploit closed-loop including standard PID controllers, due to their simple structure and parameter tuning [20]. The control law depends on the tracking error $e(t)$ defined by the difference between the desired and the measured output signals, i.e., $e(t) = r(t) - x(t)$. This signal is injected into the controlled process after Proportional, Integral and Derivative (PID) computations. Therefore, the continuous-time control signal $u(t)$ generated by the PID regulator has the form of Equation (5):

$$u(t) = K_p \, e(t) + K_i \int_0^t e(\tau) \, d\tau + K_d \frac{de(t)}{dt}, \tag{5}$$

with $K_p$, $K_i$, $K_d$ being the PID proportional, integral, and derivative gains, respectively. The most common strategy exploited for the computation of the parameters of the PID governor relies on the relations of Ziegler–Nichols [20]. However, with the development of relatively recent automatic software routines, the optimal parameters of the PID regulator can be easily determined by means of the tuning algorithm implemented in the Simulink environment. This strategy requires the implementation of the controlled process by means of the Simulink functional blocks, since it tries to optimise the input–output performances of the monitored system in terms of response time and stability margins (robustness) [20]. In particular, the automatic tuning procedure implemented by the PID Simulink block performs the computation of the linearised model of the energy conversion systems studied in this paper. The logic scheme of this procedure is sketched in Figure 5.

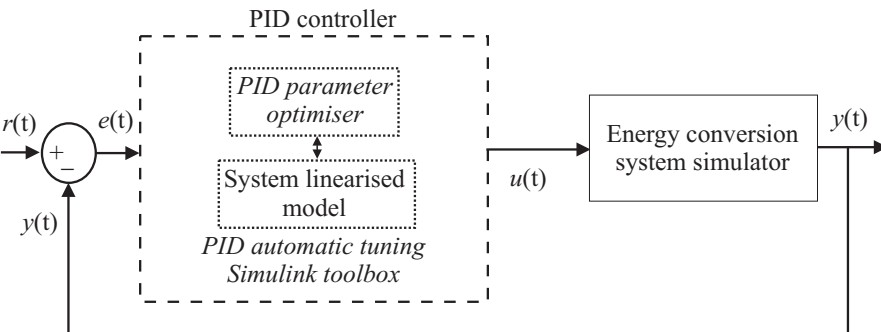

**Figure 5.** Block diagram of the monitored system controlled by the PID regulator with self-tuning feature.

According to Figure 5, the PID block performs the computation of a linearised model of the controlled system, if required. Therefore, the optimiser included in the PID block and implemented in the Simulink environment derives of the PID parameters that minimise suitable performance indices, as described in [20].

### 3.2. Data-Driven Fuzzy Control

Fuzzy Logic Control (FLC) solutions are often exploited when the dynamics of the monitored process are uncertain and it can present nonlinear characteristics. The design method proposed in this work exploits the direct identification of rule-based Takagi–Sugeno (TS) fuzzy prototypes. Moreover, the fuzzy model structure, i.e., the number of rules, the antecedents, the consequents and the fuzzy membership functions are estimated by means of the Adaptive Neuro-Fuzzy Inference System (ANFIS) toolbox implemented in the Simulink environment (R2018b, MathWorks®, Natick, MA, USA) [21].

The TS fuzzy prototype relies on a number of rules $R_i$, whose consequents are deterministic functions $f_i(\cdot)$ in the form of Equation (6):

$$R_i : IF\ x\ is\ A_i\ THEN\ u_i = f_i(x) \tag{6}$$

where the index $i = 1, 2, \ldots, K$ describes the number of rules $K$, $x$ is the input vector containing the antecedent variables, i.e., the model inputs, whilst $u_i$ represents the consequent output. The fuzzy set $A_i$ describing the antecedents in the $i$-th rule is described by its (multivariable) membership function $\mu_{A_i}(x) \rightarrow [0, 1]$. The relation $f_i(x)$ assumes the form of parametric affine model represented by Equation (7):

$$u_i = a_i^T x + b_i, \tag{7}$$

the vector $a_i$ and the scalar $b_i$ being the parameters of the $i$-th submodel. The vector $x$ consists of a proper number $n$ of delayed samples of input and output signals acquired from the monitored process. Therefore, the term $a_i^T x$ is an Auto-Regressive eXogenous (ARX) parametric dynamic model of order $n$, and $b_i$ a bias.

The output $u$ of the TS fuzzy prototype is computed as weighted average of all rule outputs $u_i$ in the form of Equation (8):

$$u = \frac{\sum_{i=1}^{K} \mu_{A_i}(x)\, y_i(x)}{\sum_{i=1}^{K} \mu_{A_i}(x)}. \tag{8}$$

The estimation scheme implemented by the ANFIS tool follows the classic dynamic system identification experiment. First, the structure of the TS fuzzy prototype is defined by selecting a suitable order $n$, the shape representing the membership functions $\mu_{A_i}$, and the proper number of clusters $K$. Therefore, the input–output data sequences acquired from the monitored system are exploited by ANFIS for estimating the TS model parameters and its rules $R_i$ after the selection of a suitable error criterion. The optimal values of the controller parameters represented by the variables $a_i$ and $b_i$ of the TS model of Equation (7) are thus estimated [21].

This work proposes also a strategy different from ANFIS that is exploited for the estimation of the parameters of the TS fuzzy model. This method relies on the Fuzzy Modelling and Identification (FMID) toolbox designed in the Matlab and Simulink environments as described in [22]. Again, the computation of the controller model is performed by estimating the rule-based fuzzy system in the form of Equation (8) from the input–output data acquired from the process under investigation. In particular, the FMID tool uses the Gustafson–Kessel (GK) clustering method [22] to perform a partition of the input–output data into a proper number $K$ of regions (clusters) where the local affine relations of Equation (7) are valid. In addition in this case, the fuzzy controller model of Equation (8) is computed after the selection of the model order $n$ and the number of clusters $K$. The FMID toolbox derives the variables $a_i$ and $b_i$, as well as the identification of the shape of the functions $\mu_{A_i}$ by minimising a given metric [22].

The overall digital control scheme consisting of the discrete-time fuzzy regulator of Equation (8) and the controlled system includes also Digital-to-Analog (D/A) and Analog-to-Digital (A/D) converters, as shown in Figure 6.

Figure 6 highlights that the fuzzy controller block implemented in the Simulink environment includes a suitable number $n$ of delayed samples of the signals acquired from the monitored process. Moreover, the fuzzy inference system in Figure 6 implements the TS model of Equation (8). The delay $n$, the membership functions $\mu_{A_i}$, and the number of clusters $K$ are estimated by the FMID and the ANFIS toolboxes, as described in [21,22].

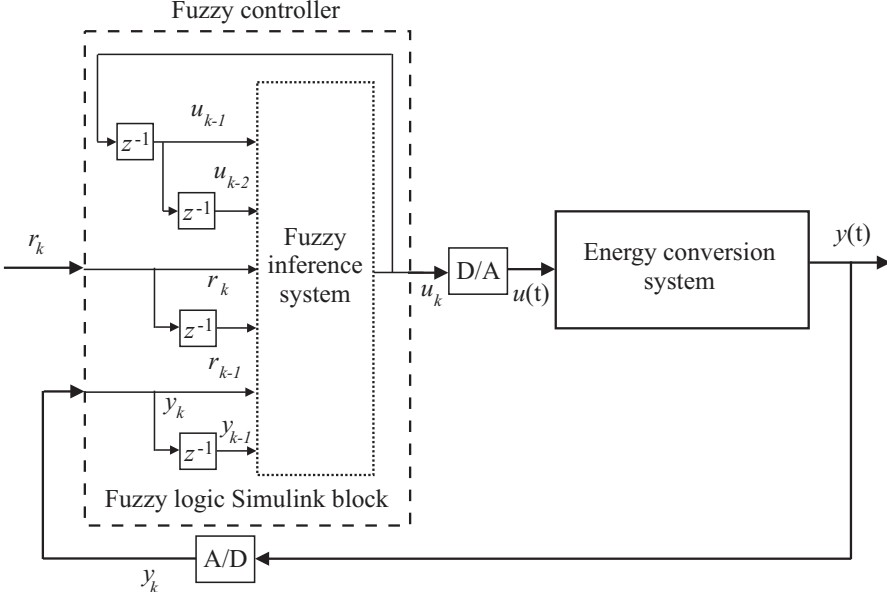

**Figure 6.** Block diagram of the monitored system controlled by the fuzzy regulator.

### 3.3. Data-Driven Adaptive Control

The adaptive control technique proposed in this work relies on the recursive estimation of a discrete-time second order transfer function $G(z)$ with time-varying parameters in the form of Equation (9):

$$G(z) = \frac{\beta_1 z^{-1} + \beta_2 z^{-2}}{1 + \alpha_1 z^{-1} + \alpha_2 z^{-2}},$$ (9)

where $\alpha_i$ and $\beta_i$ are identified online at each sampling time $t_k = k T$, with $k = 1, 2, \ldots, N$, for $N$ samples, and $T$ being the sampling interval. $z^{-1}$ indicates the unit delay operator.

This work proposes deriving the model parameters in Equation (9) by means of the Recursive Least-Square Method (RLSM) with directional forgetting factor, which was presented in [23]. Once the parameters of the model of Equation (9) have been derived, this paper suggests implementing the adaptive controller of Equation (10):

$$u_k = q_0 e_k + q_1 e_{k-1} + q_2 e_{k-2} + (1 - \gamma) u_{k-1} + \gamma u_{k-2},$$ (10)

where $e_k$ and $u_k$ represent the sampled values of the tracking error $e(t)$ and the control signal $u_k$ at the time $t_k$, respectively. With reference to the description of Equation (10), by following a modified Ziegler–Nichols criterion, the variables $q_0$, $q_1$, $q_2$, and $\gamma$ represent the adaptive controller parameters, which are derived by solving a Diophantine equation. As described in [23], by considering the second order model of Equation (9), this procedure leads to the relations of Equation (11):

$$\begin{cases} q_0 &= \frac{1}{\beta_1} \left( d_1 + 1 - \alpha_1 - \gamma \right), \\[2mm] \gamma &= \frac{s_1}{r_1} \frac{\beta_2}{\alpha_2}, \\[2mm] q_1 &= \frac{\alpha_2}{\beta_2} - \frac{s_1}{r_1} \left( \frac{\beta_1}{\beta_2} - \frac{\alpha_1}{\alpha_2} + 1 \right), \\[2mm] q_2 &= \frac{s_1}{r_1}, \end{cases}$$ (11)

where:

$$\begin{cases} r_1 &= (b_1 + b_2)\,(a_1\,b_2\,b_1 - a_2\,b_1^2 - b_2^2)\,, \\[2mm] s_1 &= a_2\,((b_1 + b_2)\,(a_1\,b_2 - a_2\,b_1) + b_2\,(b_1\,d_2 - b_2\,d_1 - b_2))\,. \end{cases} \tag{12}$$

The design technique represented by the relations of Equations (11) and (12) assumes that the behaviour of the overall closed-loop system can be approximated by a second order transfer function with characteristic polynomial represented by Equation (13):

$$D(s) = s^2 + 2\,\delta\,\omega\,s + \omega^2 \tag{13}$$

with $\delta$ and $\omega$ being the damping factor and natural frequency, respectively. $s$ is the derivative operator. Furthermore, if $\delta \leq 1$, the following relations are used [23]:

$$\begin{cases} d_1 &= -2\,e^{-\delta\,\omega\,T}\,\cos\left(\omega\,T\sqrt{1-\delta^2}\right), \\[2mm] d_2 &= e^{-2\,\delta\,\omega\,T}. \end{cases} \tag{14}$$

The online control law of Equation (10) is exploited for the regulation of the continuous-time nonlinear system by including D/A and A/D converters, as highlighted in the scheme of Figure 7.

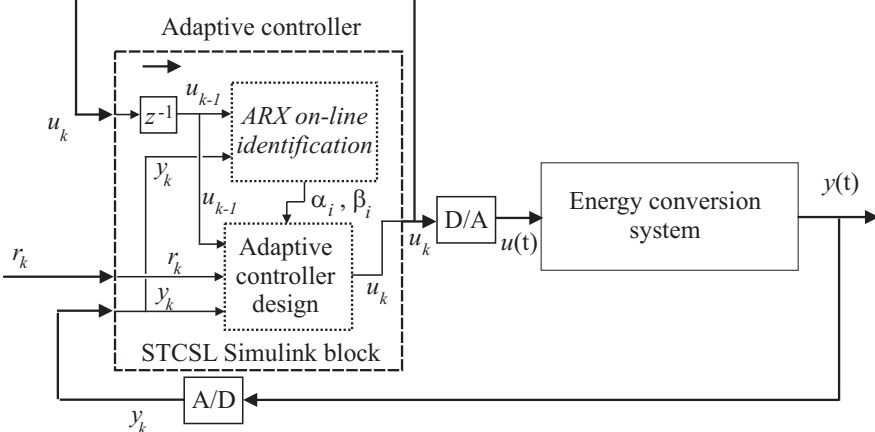

**Figure 7.** Block diagram of the monitored system controlled by the adaptive regulator.

The adaptive control sketched in Figure 7 is implemented via the Self-Tuning Controller Simulink Library (STCSL) block in the Simulink environment. It includes the module performing the online identification of the ARX model of Equation (9), which is used for the design of the adaptive Equation (10) [23].

### 3.4. Model Predictive Control with Disturbance Decoupling

The general structure of the proposed Model Predictive Control (MPC) scheme is illustrated in Figure 8. This scheme works as a standard MPC controller when the nominal plant is considered, and generates the reference inputs, by taking into account objectives and constraints. However, in the presence of disturbance or uncertainty effects, the considered solution provides the reconstruction of the equivalent disturbance signal acting on the plant. This represents the key feature of this structure, which compensates the disturbance effect, thus 'hiding' it to the overall system. In this way, it decouples the nominal MPC design from the disturbance effect.

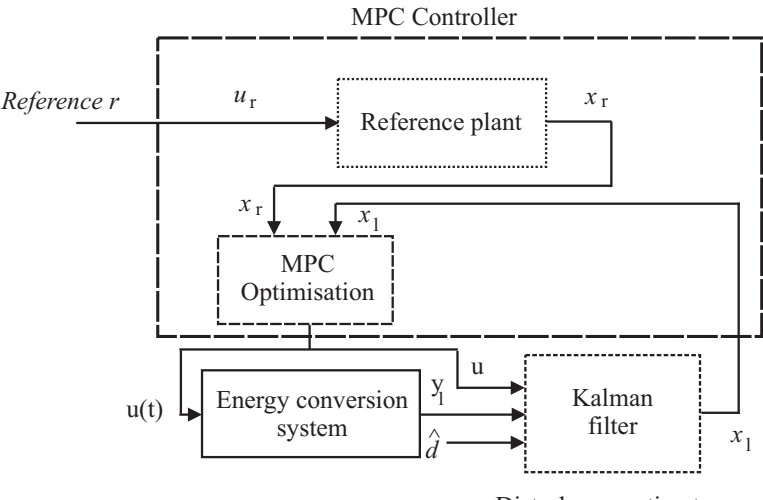

**Figure 8.** Block diagram of the disturbance compensated MPC scheme.

The complete scheme is thus represented by the MPC design that includes the disturbance compensation module, such that the compensated system has a response very similar to the nominal system and the constraints are not violated.

The disturbance compensation problem within the MPC framework is defined as follows. It is assumed that a state-space representation of the considered system is available, affected by disturbance and uncertainty. This formulation can be derived by nonlinear model linearisation or identification procedures, as suggested in Sections 3.1 and 3.3, respectively.

On the other hand, its nominal reference model has the form of Equation (15):

$$\begin{cases} \dot{x}_r &= A_l\, x_r + B_l\, u_r, \\ y_r &= C_l\, x_r. \end{cases} \tag{15}$$

The disturbance compensation problem is solved by finding the control input $u$ that minimises the cost function of Equation (16):

$$J = \int_t^{t+N_c\,\Delta t} \left( \left\| x_l - x_r \right\|_Q^2 + \left\| \dot{u} \right\|_R^2 \right) d\tau \tag{16}$$

given the reference input $u_r$.

The matrices $A_l$, $B_l$, $B_d$ and $C_l$ are of proper dimensions. $x_l$ is the state of the model with disturbance, whilst $x_r$ is the reference state, and $y_r$ the reference output, corresponding to the reference inputs $u_r$ and the output measurements $y_l$ of the nominal model.

The terms $w$ and $v$ represent the model-reality mismatch and the measurement error, respectively. $d$ is the equivalent disturbance signal. In Equation (16), $t$ is the current time, $\Delta t$ is the control interval, and $N_c$ is the length of the control horizon. $Q$ and $R$ are suitable weighting matrices.

This work proposes solving the problem in two steps. First, the reconstruction of the disturbance $d$, i.e., $\hat{d}$, is provided by the disturbance estimation module. Then, the MPC design is executed. Due to the model-reality mismatch and the measurement error in the representation of Equation (17):

$$\begin{cases} \dot{x}_l &= A_l\, x_l + B_l\, u + B_d\, d + w, \\ y_l &= C_l\, x_l + v, \end{cases} \tag{17}$$

the Kalman filter of Equation (18) is exploited to provide the estimation of the state vector $x_l$ and the output $y_l$ of the system affected by the estimated disturbance $\hat{d}$:

$$\begin{cases} \dot{x}_l &= A_l\, x_l + B_l\, u - B_l\, \hat{d} + K_f\, (y_l - C_l\, x_l)\,, \\ y_l &= C_l\, x_l, \end{cases} \tag{18}$$

where $K_f$ is the Kalman filter gain. In this way, based on the estimations $\hat{d}$ and $x_l$, the MPC with disturbance compensation is designed, which consists of the model of Equation (17) and the system of Equation (18), with $\hat{d}$ provided by the Kalman filter. Moreover, the MPC has the objective function of Equation (19):

$$\int_{t}^{t+N_c\,\Delta t} \left[ (x_l - x_r)^T\, Q\, (x_l - x_r) + \dot{u}^T R\, \dot{u} \right]\, d\tau \tag{19}$$

in which $x_l$ and $x_r$ are the states of the filtered and the reference models, respectively. The MPC scheme including the Kalman filter solves the disturbance compensation problem, as long as the estimations of both the state and the disturbance terms are correct. An illustration of the structure of the disturbance compensated MPC is shown in Figure 8.

The proposed technique leads to a nonlinear MPC problem that includes the nominal model of the considered energy conversion system of Equation (17), the estimator of the disturbance $d$, and the Kalman filter of Equation (18) as predictor. The local observability of the model of Equation (17) is essential for state estimation, which is easily verified. The implementation of the proposed disturbance compensation strategy has been integrated into the MPC Toolbox of the Simulink environment.

## 4. Simulation Results

The results obtained from the application of the developed control techniques are evaluated via the percent Normalised Sum of Squared Error (*NSSE*%) performance index in the form of Equation (20):

$$NSSE\% = 100\, \sqrt{\frac{\sum_{k=1}^{N} (r_k - o_k)^2}{\sum_{k=1}^{N} r_k^2}} \tag{20}$$

with $r_k$ being the sampled reference or set-point $r(t)$, whilst $o_k$ is the sampled continuous-time signal representing the generic controlled output $y(t)$ of the process. In particular, this signal is represented by the wind turbine generator angular velocity $\omega_g(t)$ in Equation (2), and the hydraulic turbine rotational speed $n$ in Equation (3) for the hydroelectric plant.

Note that the wind turbine benchmark and the hydroelectric plant simulator of Section 2 allow the generation of several input–output data sequences driven by different wind speed $v(t)$ processes and hydraulic transient under variable loads, respectively. Moreover, in order to obtain comparable working situations, the wind turbine benchmark operates from partial to full load conditions (from region 2 to region 3). It is thus considered the similar maneuver of the hydroelectric system operating from the start-up to full load working conditions. After these considerations, Section 4.1 summarises the results obtained from the wind turbine benchmark. Then, the same control techniques will be verified when applied to the hydroelectric simulator.

### 4.1. Control Technique Performances and Comparisons

This section reports the results achieved from the application of the control techniques and the tools summarised in Section 3 to the wind turbine and the hydroelectric simulators recalled in Section 2.

In particular, Figure 9 depicts the wind turbine generator angular velocity $\omega_g$ when the wind speed $v(t)$ changes from 3 m/s to 18 m/s for a simulation time of 4400 s [6].

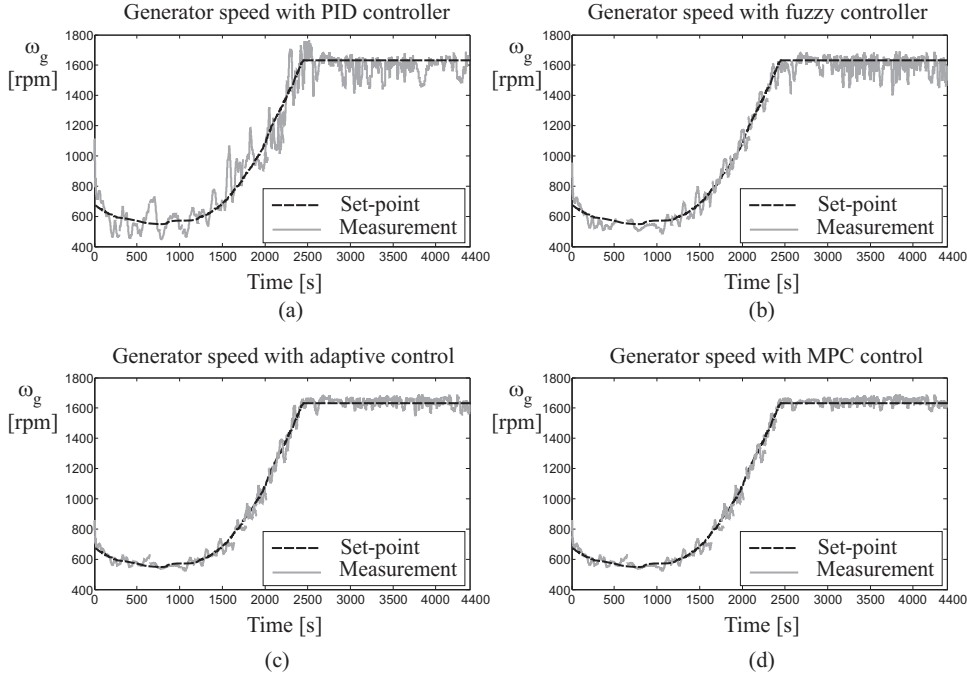

**Figure 9.** Wind turbine controlled output compensated by (**a**) the self-tuning PID regulator, (**b**) the fuzzy controller, (**c**) the adaptive regulator, and (**d**) the MPC approach with disturbance decoupling.

With reference to Figure 9a, the parameters of the PID regulator of Equation (5) have been determined using the self-tuning tool available in the Simulink environment. They were settled to $K_p = 4.0234$, $K_i = 1.0236$, and $K_d = 0.0127$. The achieved performances are better than those obtained with the baseline control law developed in [6].

Moreover, Figure 9b shows the simulations achieved with the data-driven fuzzy identification approach of Section 3.2. A sampling interval $T = 0.01$ s has been exploited, and the TS fuzzy controller of Equation (8) has been obtained for a number $K = 3$ of Gaussian membership functions, and a number $n = 2$ of delayed inputs and output. Therefore, the antecedent vector in Equation (7) is $x = [e_k, e_{k-1}, e_{k-2}, u_{k-1}, u_{k-2}]$. Both the data-driven FMID and ANFIS tools available in the Matlab and Simulink environments provide also the identification of the shapes of the fuzzy membership functions $\mu_{A_i}$ of the fuzzy sets $A_i$ in Equation (6).

On the other hand, Figure 9c shows the capabilities of the adaptive controller of Equation (10). The time-varying parameters of this data-driven control technique summarised in Section 3.3 have been computed online via the relations of Equation (11) with the damping factor and the natural frequency variables $\delta = \omega = 1$ in Equation (13).

Finally, Figure 9d highlights the results achieved with the MPC technique illustrated in Section 3.4. A state-space model with $n = 5$ in Equation (2) of the wind turbine nonlinear system is exploited to design the MPC and the Kalman filter for the estimation of the disturbance, with a prediction horizon $N_p = 10$ and a control horizon $N_c = 2$. The weighting factors have been settled to $w_{y_k} = 0.1$ and $w_{u_k} = 1$, in order to reduce possible abrupt changes of the control input. In this case, the MPC technique has led to the best results, since it exploits a disturbance decoupling strategy, whilst its parameters have been iteratively adapted in the Simulink environment in order to optimise the MPC cost function of Equation (16), as addressed in Section 3.4.

The second test case concerns the hydroelectric plant simulator, where the hydraulic system with its turbine speed governor generates hydraulic transients due to the load changes. In order to consider operating situations similar to those of the wind turbine benchmark, the capabilities of the considered control techniques applied to the hydroelectric simulator have been evaluated during the start-up to full load maneuvers. To this end, an increasing load torque $m_{g0}$ has been imposed during the start-up

to full load phase, which is assumed to last 300 s, because of the large size of the considered Francis turbine, and for a simulation of 900 s.

Under these assumptions, Figure 10 summarises the results achieved with the application of the control strategies recalled in Section 3. In particular, for all cases, Figure 10 highlights that the hydraulic turbine angular velocity $n$ increases with the load torque $m_{g0}$ during the start-up to full working condition maneuver.

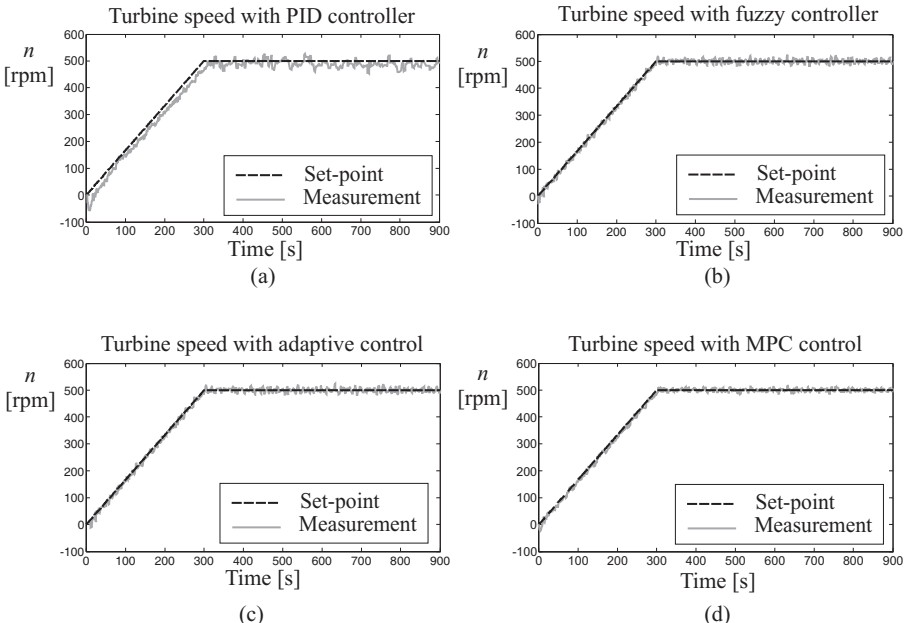

**Figure 10.** Hydroelectric system with (**a**) the self-tuning PID regulator, (**b**) the fuzzy controller, (**c**) the adaptive regulator, and (**d**) the MPC approach with disturbance decoupling.

In more detail, Figure 10a shows the performance of the PID regulator whose parameters are determined via the self-tuning procedure recalled in Section 3.1. Furthermore, Figure 10a shows that the PID governor with self-tuning capabilities is able to keep the hydraulic turbine rotational speed error $n - n_r$ null ($r(t) = n_r$, i.e., the rotational speed constant) in steady-state conditions.

Figure 10b reports the results of the TS fuzzy controller of Equation (8). This fuzzy controller was implemented for a sampling interval $T$ = 0.1 s, with $K$ = 2 Gaussian membership functions, and $n$ = 3 delayed inputs and output. Therefore, the antecedent vector exploited by the relation of Equation (7) is $x = [e_k, e_{k-1}, e_{k-2}, e_{k-3}, u_{k-1}, u_{k-2}, u_{k-3},]$. Moreover, as recalled in Section 3.2, the data-driven FMID and ANFIS tools implemented in the Simulink toolboxes are able to provide the estimates of the shapes of the membership functions $\mu_{A_i}$ used in Equation (8).

On the other hand, Figure 10c reports the simulations obtained via the data-driven adaptive controller of Equation (10), whose time-varying parameters are computed by means of the relations of Equation (11). The damping factor and the natural frequency parameters used in Equation (13) were fixed to $\delta = \omega = 1$. The STCSL tool recalled in Section 3.3 implements this data-driven adaptive technique using the online identification of the input–output model of Equation (9) [23].

Finally, regarding the MPC technique with disturbance decoupling proposed in Section 3.4, Figure 10d reports the simulations obtained using a prediction horizon $N_p$ = 10 and a control horizon $N_c$ = 2. In addition, in this case, the weighting parameters have been fixed to $w_{y_k}$ = 0.1 and $w_{u_k}$ = 1, in order to limit fast variations of the control input, as it will be remarked in the following. Furthermore, the MPC design was performed using a linear state-space model of order $n$ = 6 for the nonlinear hydroelectric plant simulator of Equation (3).

In order to provide a quantitative comparison of the tracking capabilities obtained by the considered control techniques for the wind turbine benchmark, Table 1 summarises the achieved results in terms of *NSSE*% index.

**Table 1.** Performance of the considered control solutions for the wind turbine.

| Simulated System | Working Condition | Standard PID | Self-Tuning PID | Fuzzy PID | Adaptive PID | MPC Scheme |
|---|---|---|---|---|---|---|
| Wind turbine | From partial to full load | 11.5% | 7.3% | 5.7% | 4.1% | 2.8% |

In particular, the *NSSE*% values in Table 1 highlight that the fuzzy controllers lead to better performances than the PID regulators with self-tuning features. This is motivated by the flexibility and the generalisation capabilities of the fuzzy tool, and in particular the FMID toolbox proposed in [22]. Better results are obtained by means of the adaptive solution, due to its inherent adaptation mechanism, which allows for tracking the reference signal in the different working conditions of the wind turbine process. However, the MPC technique with disturbance decoupling has achieved the best results, as reported in Table 1, since it is able to optimise the overall control law over the operating conditions of the system, by taking into account future operating situations of its behaviour, while compensating the disturbance effects.

On the other hand, the results achieved by the application of the considered control techniques to the hydroelectric plant simulator are summarised in Table 2.

**Table 2.** Performance of the considered control solutions for the hydroelectric plant.

| Simulated System | Working Condition | Standard PID | Self-Tuning PID | Fuzzy PID | Adaptive PID | MPC Scheme |
|---|---|---|---|---|---|---|
| Hydro plant | From start-up to full load | 6.2% | 4.9% | 3.1% | 1.8% | 0.9% |

In this case, the values of the *NSSE*% index are evaluated for the considered conditions of varying load torque $m_{g0}$ from the plant start-up to the full load maneuver. According to these results, good properties of the proposed self-tuning PID regulator are obtained, and they are better than the baseline PID governor with fixed gains developed in [19]. In fact, the self-tuning design feature of the Simulink environment is able to limit the effect of high-gains for the proportional and the integral contributions of the standard PID control law. On the other hand, the data-driven fuzzy regulator has led to even better results, which are outperformed by the adaptive solution. However, also for the case of the hydroelectric plant simulator, the best performances are obtained by means of the MPC strategy with disturbance decoupling.

Finally, in order to highlight some further characteristics of the developed control strategies, the actuated inputs $\beta(t)$ and $\tau_r(t)$ feeding the wind turbine system are depicted in Figure 11, i.e., the blade pitch angle and the generator reference torque. On the other hand, Figure 12 depicts the control input $u$ of the hydraulic turbine of the hydroelectric plant. For the sake of brevity, only the results for the data-driven fuzzy controller and the MPC with disturbance decoupling have been reported.

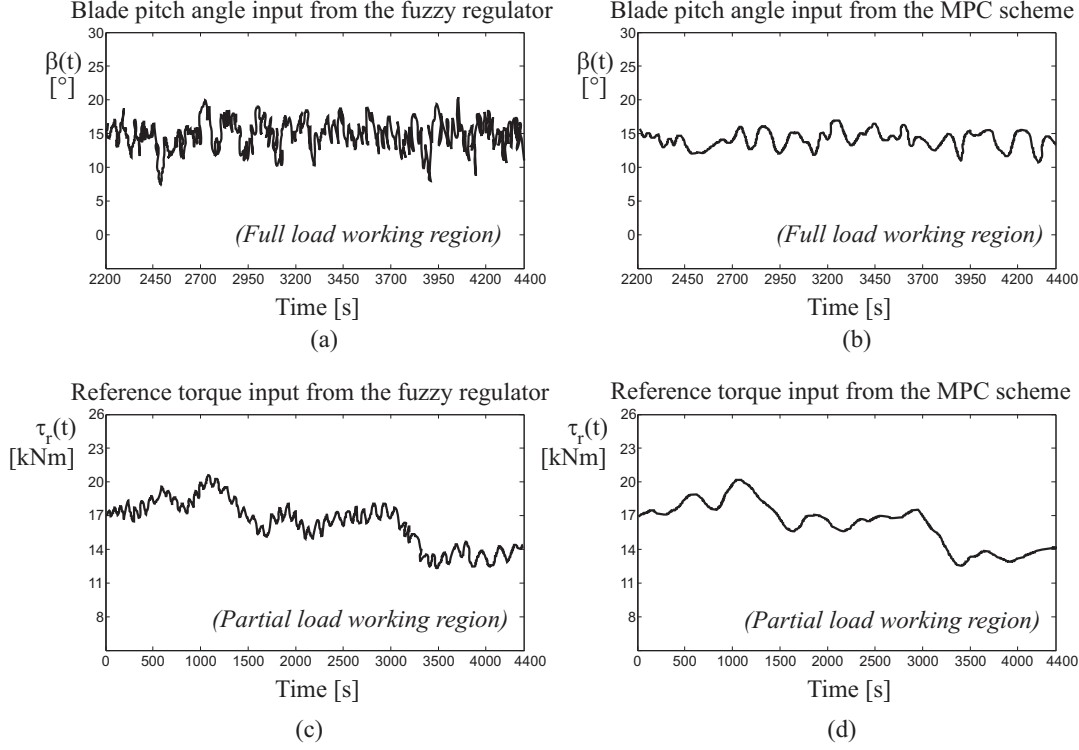

**Figure 11.** Wind turbine inputs (**a**,**c**) from the fuzzy control strategy and (**b**,**d**) by the MPC scheme.

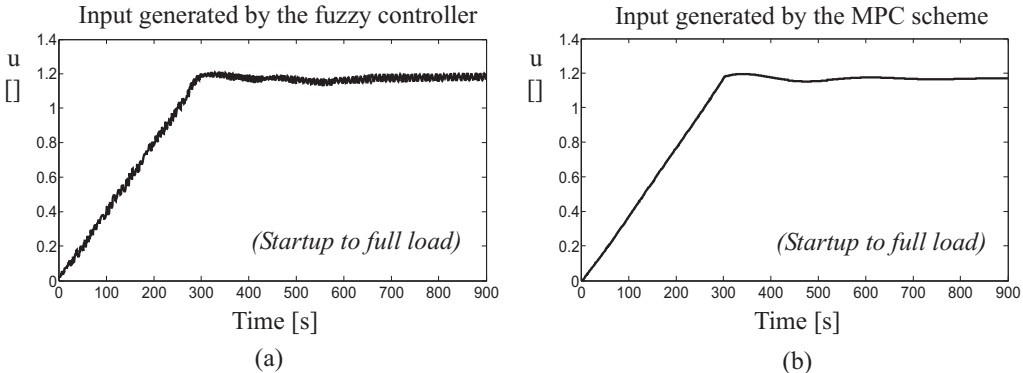

**Figure 12.** Hydroelectric plant input *u* generated (**a**) by the fuzzy controller and (**b**) from the MPC approach.

By considering these control inputs, with reference to the data-driven methodologies, and in particular to the design of the fuzzy controllers, offline optimisation strategies allow to reach quite good results. However, control inputs are subjected to faster variations, as shown in Figure 11a,c, and Figure 12a. Other control techniques take advantage of more complicated and not direct design methodologies, as highlighted by the MPC scheme. In this case, due to the input constraint, its changes are reduced, as shown in Figure 11b,d, and Figure 12b. This feature is attractive for wind turbine systems, where variations of the control inputs must be reduced. This represents another important benefit of MPC with disturbance decoupling, which integrates the advantages of the classic MPC scheme with disturbance compensation capabilities. Therefore, with reference to these two control methods, they can appear rather straightforward, even if further optimisation and estimation strategies have to be applied.

### 4.2. Sensitivity Analysis

This section analyses the reliability and robustness properties of the developed controllers when the simulations include parameter variations and measurement errors. This further investigation exploits the Monte Carlo tool, since the control behaviour and the tracking capabilities depend on both the model-reality mismatch effects and the input–output error levels. Therefore, this analysis has been implemented by describing the parameters of both the wind turbine system and hydroelectric plant models as Gaussian stochastic processes. Their average values corresponding to the nominal ones are summarised in Table 3 for the wind turbine benchmark.

**Table 3.** Wind turbine benchmark parameters for the sensitivity analysis.

| Variable | $R$ | $\chi$ | $\omega_n$ | $B_{dt}$ | $B_r$ |
|---|---|---|---|---|---|
| Nominal value | 57.5 m | 0.6 | 106.09 rpm | 775.49 N m s rad$^{-1}$ | 7.11 N m s rad$^{-1}$ |
| Variable | $B_g$ | $K_{dt}$ | $\eta_{dt}$ | $J_g$ | $J_r$ |
| Nominal value | 45.6 N m s rad$^{-1}$ | $2.7 \cdot 10^9$ N m rad$^{-1}$ | 0.97 | 390 kg m$^2$ | $55 \cdot 10^6$ kg m$^2$ |

Moreover, Table 3 shows that these model parameters have standard deviations of $\pm 30\%$ of the corresponding nominal values [6].

On the other hand, Table 4 reports the hydroelectric simulator model variables with their nominal values varied by $\pm 30\%$ in order to execute the same Monte Carlo analysis [7].

**Table 4.** Hydroelectric simulator parameters for the sensitivity analysis.

| Variable | $a$ | $b$ | $c$ | $H_{f_1}$ | $H_{f_3}$ | $H_{f_5}$ | $T_a$ |
|---|---|---|---|---|---|---|---|
| Nominal value | $-0.08$ | 0.14 | 0.94 | 0.0481 m | 0.0481 m | 0.0047 m | 5.9 s |
| Variable | $T_c$ | $T_{s_2}$ | $T_{s_4}$ | $T_{w_1}$ | $T_{w_3}$ | $T_{w_5}$ | |
| Nominal value | 20 s | 476.05 s | 5000 s | 3.22 s | 0.83 s | 0.1 s | |

Therefore, the average values of $NSSE\%$ index have been thus evaluated by means of 1000 Monte Carlo simulations. They have been reported in Tables 5 and 6 for the wind turbine benchmark and the hydroelectric plant simulator, respectively.

**Table 5.** Sensitivity analysis applied to the wind turbine benchmark.

| Standard PID | Self-Tuning PID | Fuzzy PID | Adaptive PID | MPC Scheme |
|---|---|---|---|---|
| 13.8% | 9.2% | 7.6% | 5.3% | 3.9% |

**Table 6.** Sensitivity analysis applied to the hydroelectric plant simulator.

| Standard PID | Self-Tuning PID | Fuzzy PID | Adaptive PID | MPC Scheme |
|---|---|---|---|---|
| 9.1% | 7.4% | 5.6% | 3.5% | 2.2% |

It is worth noting that the results summarised in Tables 5 and 6 serve to assess the overall behaviour of the developed control techniques. In more detail, the values of the $NSSE\%$ index highlights that when the mathematical description of the controlled dynamic processes may be included in the control design phase, the MPC technique with disturbance decoupling still yields to the best performances, even if an optimisation procedure is required. However, when modelling errors are present, the offline learning feature of the data-driven fuzzy regulators allows for achieving better results than model-based schemes. For example, this consideration is valid for the PID controllers

derived via the self-tuning procedure. On the other hand, fuzzy controllers have led to interesting tracking capabilities. With reference to the adaptive scheme, it takes advantage of its recursive features, since it is able to track possible variations of the controlled systems, due to operation or model changes. However, it requires quite complicated and not straightforward design procedures relying on data-driven recursive algorithms. Therefore, fuzzy-based schemes use the learning accumulated from data-driven offline simulations, but the training stage can be computationally heavy. Finally, concerning the standard PID control strategy, which represented the baseline regulator for the considered processes, it is rather simple and straightforward. Obviously, the achievable performances are quite limited when applied to nonlinear dynamic processes. It can thus be concluded that the proposed data-driven self-tuning approaches seem to represent powerful techniques able to cope with uncertainty, disturbance and variable working conditions. The plant simulators, the control solutions, and the data exploited for the analysis addressed in this paper are directly and freely available from the authors.

## 5. Conclusions

The work considered two renewable energy conversion systems, namely a wind turbine benchmark and a hydroelectric plant simulator, together with the development of proper data-driven control techniques. In particular, the three-bladed horizontal axis wind turbine benchmark reported in this work consisted of simple models of the gear-box, the drive-train, and the electric generator/converter. On the other hand, the hydroelectric plant simulator included a high water head, a long penstock with upstream and downstream surge tanks, and a Francis hydraulic turbine. Standard PID governors were earlier developed for these processes, which were rather simple and straightforward, but with limited achievable performances. Therefore, the paper proposed advanced control strategies mainly relying on data-driven approaches. Their performances were analysed first. Then, the reliability and robustness of these solutions were also verified and validated with respect to parameter variations of the plant models and measurement errors via the Monte Carlo tool. The achieved results highlighted that data-driven approaches, such as the fuzzy regulators were able to provide good tracking performances. However, they were easily outperformed by adaptive and model predictive control schemes, representing data-driven solutions that require optimisation stages, adaptation procedures and disturbance compensation methods. Future investigations will consider the verification and the validation of the considered control techniques when applied to higher fidelity simulators of energy conversion systems.

**Author Contributions:** S.S. conceived and designed the simulations. S.S. analysed the methodologies, the achieved results, and, together with S.A. and M.V., wrote the paper.

**Funding:** This research received no external funding.

**Acknowledgments:** The costs to publish in open access have been covered by the FIR2018 local fund from the University of Ferrara.

**Conflicts of Interest:** The authors declare no conflict of interest.

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
