# Peer review of "Data-Driven Control Techniques for Renewable Energy Conversion Systems: Wind Turbine and Hydroelectric Plants"

_electronics, doi:10.3390/electronics8020237_

Round 1
Reviewer 1 Report
The original manuscript has been improved, but some changes are needed:
1. Avoid using “which is not addressed in this paper” (line 55)
2. Highlight the main idea behind “More details on this simulator are available in [6]” (line 86) or delete this sentence.
3. Explain better lines 119-121.
4. Highlight the main idea behind “More details can be found in [7]” (line 121) or delete this sentence.
5. Sections 3.0.1, 3.0.2, 3.0.3 and 3.0.4 should be called 3.1, 3.2, 3.3 and 3.4, respectively.
6. What does “Note that some of the control solutions were already proposed by the authors in particular for the wind turbine and wind park installations [20]” (lines 171-172) mean? It should be clarified.
7. There are too many “Note…”
8. Eq. (15) and Eq. (18) should be written properly in the first paragraph in page 11.
9. In lines 281, 292 and 296 “the picture in Figure x” should be replaced with “Figure x”.
10. In page 14, can the authors write “the first row in Table 1” in a different way?
11. Rewrite Figure 11 title: the cases are unclear.
12. In Table 2, 106.09 rpm should be written.
Author Response
The paper has been revised according to the comments received by the two anonymous referees. All changes have been highlighted in yellow in the revised version of the paper and the responses to reviewers’ comments are reported in the following.
The authors would like to thank the Managing Editor and the anonymous referee for her/his comments and suggestions, which were taken into account for improving the revised version of the manuscript. In the following, the anonymous referee’s comments and remarks are summarised (in black colour text) and followed by the authors’ answers, highlighted in blue colour.
Anonymous Referee #1 Comments and Authors’ Answers
The original manuscript has been improved, but some changes are needed.
The authors would like to thank the anonymous referee for this remark. In the following, the referee’s comments have been properly answered and commented.
1. Avoid using “which is not addressed in this paper” (line 55)
The authors would like to thank the anonymous referee for this remark. These occurrences have been removed in the revised version of the paper.
2. Highlight the main idea behind “More details on this simulator are available in [6]” (line 86) or delete this sentence.
The authors would like to thank the anonymous referee for this remark. The sections addressing the details of the considered simulators were considerably reduced, as suggested by the referees in the previous round of reviewing. However, the authors agree that the remarked sentence can be safely removed.
3. Explain better lines 119-121.
The authors would like to thank the anonymous referee for this remark. The sentence means that [17, 18] describe the hydraulic circuit which does not include the Francis turbine. In fact, the simple hydraulic system was available in the literature [17, 18] for the analysis of hydraulic transients. However, the authors modified this simple hydraulic circuit in order to include a Francis turbine that motivated the use of this simulator as hydroelectric plant. This sentence has been modified in the revised version of the paper.
4. Highlight the main idea behind “More details can be found in [7]” (line 121) or delete this sentence.
The authors would like to thank the anonymous referee for this remark. As already commented at point 2, the sections addressing the details of the considered simulators were considerably reduced, as suggested by the referees in the previous round of reviewing. However, the authors agree that the remarked sentence can be safely removed.
5. Sections 3.0.1, 3.0.2, 3.0.3 and 3.0.4 should be called 3.1, 3.2, 3.3 and 3.4, respectively.
The authors would like to thank the anonymous referee for this remark. This issue has been solved in the revised version of the paper.
6. What does “Note that some of the control solutions were already proposed by the authors in particular for the wind turbine and wind park installations [20]” (lines 171-172) mean? It should be clarified.
The authors would like to thank the anonymous referee for this remark. The authors would like to underline that the fuzzy solutions were exploited for fault diagnosis and fault tolerant control purpose in a different application example. However, the authors agree that the sentence can be unclear, and it has been removed in the revised version of the paper.
7. There are too many “Note…”
The authors would like to thank the anonymous referee for this remark. These occurrences have been removed in the revised version of the paper.
8. Eq. (15) and Eq. (18) should be written properly in the first paragraph in page 11.
The authors would like to thank the anonymous referee for this remark. Eq. (15) and Eq. (18) have been moved in the first paragraph in page 11 of the revised version of the paper, which have thus become Eq. (17) and Eq. (18).
9. In lines 281, 292 and 296 “the picture in Figure x” should be replaced with “Figure x”.
The authors would like to thank the anonymous referee for this remark. These occurrences have been removed in the revised version of the paper.
10. In page 14, can the authors write “the first row in Table 1” in a different way?
In order to remove this sentence as required by the anonymous referee, Table 1 has been split into 2 tables (Table 1 and Table 2) in the revised version of the paper.
11. Rewrite Figure 11 title: the cases are unclear.
The authors would like to thank the anonymous referee for this remark. The cases have been rewritten in Figure 11 title in the revised version of the paper. Moreover, in order to make the picture clearer, it has been split into 2 different figures, one showing the results for the wind turbine simulator, and another one for the hydroelectric plant. A few more comments have been also added to the section in the revised version of the paper.
12. In Table 2, 106.09 rpm should be written.
The authors would like to thank the anonymous referee for this remark. This issue has been solved in the revised version of the paper.

Reviewer 2 Report
Authors has done a effort to improve the previous manuscript. However, there still are some improvable parts. I also believe that a complete grammatical revision should be made.
#1: Abstract: Please include two sentences remarking both the paper objectives and a short summary of the results.
#2: Line 37: The sentence must be written as "With reference(...), it can (...)." Please, check the grammar of the entire manuscript.
#3: Line 57: What energy conversion systems? Please indicate!
#4: Figure 2 is not clear
#5: Please, reconsider Figure 4. It is not clear the meaning of "Efficiency = 0%". Consider, for example, two Y axes (one for Q/Qr and the other for the efficiency).
Author Response
The paper has been revised according to the comments received by the two anonymous referees. All changes have been highlighted in yellow in the revised version of the paper and the responses to reviewers’ comments are reported in the following.
The authors would like to thank the Managing Editor and the anonymous referee for her/his comments and suggestions, which were taken into account for improving the revised version of the manuscript. In the following, the anonymous referee’s comments and remarks are summarised (in black colour text) and followed by the authors’ answers, highlighted in blue colour.
Anonymous Referee #2 Comments and Authors’ Answers
Authors has done a effort to improve the previous manuscript. However, there still are some improvable parts. I also believe that a complete grammatical revision should be made.
The authors would like to thank the anonymous referee for this remark. In the following, the referee’s comments have been properly answered and commented. Moreover, the English of the paper has been also revised.
#1: Abstract: Please include two sentences remarking both the paper objectives and a short summary of the results.
The authors would like to thank the anonymous referee for this remark. The authors added two sentences summarising paper’s objectives and final results. The Abstract of the revised version of the paper has been modified accordingly.
#2: Line 37: The sentence must be written as "With reference(...), it can (...)." Please, check the grammar of the entire manuscript.
The authors would like to thank the anonymous referee for this remark. The sentence has been modified in the revised version of the paper.
#3: Line 57: What energy conversion systems? Please indicate!
The authors would like to thank the anonymous referee for this remark. The sentence has been modified in the revised version of the paper.
#4: Figure 2 is not clear
The authors would like to thank the anonymous referee for this remark. Figure 2 has been modified in the revised version of the paper. The picture has been redrawn in order to make the depicted curves clearer.
#5: Please, reconsider Figure 4. It is not clear the meaning of "Efficiency = 0%". Consider, for example, two Y axes (one for Q/Qr and the other for the efficiency).
The authors would like to thank the anonymous referee for this remark. Figure 4 has been modified in the revised version of the paper. The picture has been redrawn in order to make the depicted curves clearer. Moreover, it has been depicted only for different values of the wicket gate opening, thus making Figure 4 (with different values of the wicket gate opening) closer to the corresponding Figure 2 (with different values of the blade pitch angle).

Round 2
Reviewer 2 Report
Congratulations for the work done!
This manuscript is a resubmission of an earlier submission. The following is a list of the peer review reports and author responses from that submission.
Round 1
Reviewer 1 Report
This work is poor and should be improved. The main contributions of this work should be highlighted and the novelty is not clear.
The manuscript should be reduced, rewritten and explained better. Several parts in the manuscript are redundant and others are repeated. Some parts should be summarized. Many of the concepts which are explained in the manuscript are well-known by the potential readers and can be reduced.
There are a lot of “e.g. in [Ref]”, could the key findings of those references be written?
More recent references should be added. There are too many conferences and proceedings.
Reviewer 2 Report
This is well presented paper which compares different types of control strategy for wind turbine and hydroelectric generator controls. I have a couple of minor concerns:
It would have been appropriate to report the generator /rotor speed (rotational speed in general) in rpm instead of / along with rad/sec.
In page 7, Region 1 and Region 2 are defined as the below rated wind speed operation region and rated/above rated wind speed operation region. However, I have seen many literature referring them Region 2 and Region 3 respectively. Region 1 is usually used for the idle state of wind turbine where wind speed is just sufficient to turn on the wind turbine and energy produced is about enough to cover the losses.
Reviewer 3 Report
The manuscript is too long! Sections previous simulations are too extensive (It seems a review!) In fact, please highlight contributions clearly in the introduction and abstract.
To reduce the extension, I suggest to summarize sections 1,2 and 3.
Furthermore, Figure 1 is not clear and it does not provide any information. In figure 2a, please take care with the units.
Regarding the state of art, it should be extended takeing into account some of the related papers: "Dual Frequency Regulation in Pumping Mode in a Wind–Hydro Isolated System", "Eigen analysis of wind–hydro joint frequency regulation in an isolated power system", "Generic dynamic wind turbine models for power system stability analysis: A comprehensive review". These papers, and references used in these papers can help authors to improve the state of the art.